# The Effects of Exercise during Pregnancy on Gestational Diabetes Mellitus, Preeclampsia, and Spontaneous Abortion among Healthy Women—A Systematic Review and Meta-Analysis

**DOI:** 10.3390/ijerph20126069

**Published:** 2023-06-06

**Authors:** Camilla Paludan Paulsen, Elisabeth Bandak, Henriette Edemann-Callesen, Carsten Bogh Juhl, Mina Nicole Händel

**Affiliations:** 1Department of Sports Science and Clinical Biomechanics, University of Southern Denmark, 5230 Odense, Denmark; 2The Parker Institute, Bispebjerg and Frederiksberg Hospital, 2000 Frederiksberg, Denmark; 3Department of Biomedical Sciences, University of Copenhagen, 2200 Copenhagen, Denmark; 4Metodekonsulent Callesen, 8600 Silkeborg, Denmark; 5Department Physiotherapy and Occupational Therapy, Copenhagen University Hospital, Herlev and Gentofte, 2100 Copenhagen, Denmark; 6Research Unit OPEN, Department of Clinical Research, University of Southern Denmark, 5230 Odense, Denmark

**Keywords:** pregnancy, exercise, preeclampsia, gestational diabetes mellitus, spontaneous abortion, meta-analysis

## Abstract

The aim was to compare the effects of different exercise modalities (aerobic, resistance, aerobic and resistance combined, or mind–body exercise) on gestational diabetes mellitus (GDM), preeclampsia, spontaneous abortion, withdrawal from the study, and adverse events in healthy pregnant women. A systematic search was conducted in February 2022 using MEDLINE, EMBASE, Cochrane library, and SPORT Discus to identify eligible randomized trials. The meta-analysis of 18 studies that examined exercise compared to no exercise showed a reduced risk of GDM (RR: 0.66 (95% CI: 0.50 to 0.86)). No subgroup differences were found regarding modality, intensity, or supervision. Exercise did not reduce the risk of preeclampsia (nine studies, RR: 0.65 (95% CI: 0.42 to 1.03)); however, in subgroup analyses, mind–body exercise and low-intensity exercise seemed to be effective in reduction of preeclampsia. There was no effect of exercise on withdrawal or adverse events found. No studies reported on spontaneous abortion, therefore, exercise during pregnancy is beneficial and safe. In the prevention of GDM, any modality and intensity seem equally effective. Subgroup analyses support an association between mind–body exercise and physical activity with low intensity and reduced risk of preeclampsia, but more high-quality randomized studies are needed. PROSPERO: CRD42022307053.

## 1. Introduction

Exercise during pregnancy is beneficial for both mother and child and is therefore recommended in numerous national and international guidelines [1,2,3]. The World Health Organization (WHO) recommends that healthy pregnant women undertake 150 min of weekly moderate physical activity [2]. The exercise program for pregnant women may consist of walking, cycling, swimming, running, or resistance training (in a sitting position and no heavy strength exercise) [4]. However, few pregnant women meet these recommendations for physical activity and a reduced level of exercise may occur already in early pregnancy [5].

Pregnant women often decrease their level of physical activity [6], which often is due to a combination of multiple barriers such as lack of time, lack of energy, concern about spontaneous abortion or the fetus’s wellbeing, and discomfort while exercising [7,8]. In addition, woman often experience conflicting advice about exercise from the health professionals that make them insecure regarding benefits and harms [8].

Inactivity in the prenatal period increases the risk of preeclampsia, gestational diabetes mellitus (GDM), gestational hypertension, excessive gestational weight gain, delivery complications, postpartum depression, and complications in the newborn (i.e., low birth weight, stillbirth) [2]. GDM is one of the most normal complications during pregnancy [9], and 50% of pregnant women with GDM develop type 2 diabetes mellitus within ten years after giving birth. GDM during pregnancy can increase the incidence of preeclampsia, macrosomia, and cesarean delivery, and these adverse events may affect the long-term health of both the mother and child [10]. In addition to GDM, preeclampsia affects around 5% of pregnant women worldwide and is a serious complication that often results in premature birth and long-term cardiovascular disease in the mother [11]. Therefore, preventive interventions for both GDM and preeclampsia are warranted.

Little is known on which exercise modality is the most beneficial to reduce risk of GDM, preeclampsia, and spontaneous abortion. Cordero et al. [12] state that there is a lack of specific recommendations concerning the modality of exercise intervention, including when to start, the duration, and the intensity of the exercise sessions among others. A better understanding of what influences the risk for these outcomes are needed to help the health professionals to provide the best preventive treatment for the pregnant women.

Therefore, the aim of this study was to compare the effects of different exercise modalities (aerobic, resistance, aerobic and resistance combined, or mind–body exercise) on the risk of GDM, preeclampsia, spontaneous abortion, withdrawal, and adverse events in healthy pregnant women.

## 2. Materials and Methods

This systematic review followed the Preferred Reporting Items for Systematic Reviews and Meta-analyses (PRISMA) Checklist [13], and the guidelines from the *Cochrane Handbook for Systematic Review of Interventions* [14].

### 2.1. Protocol and Registration

The protocol was registered in advance at Prospective Register of Systematic Reviews (PROSPERO) with the registration ID: CRD42022307053. The protocol was uploaded on 26 January 2022. Regarding protocol deviations, the plan was to make a network meta-analysis. However, this was not possible as none of the included studies directly compared different exercises interventions.

### 2.2. Eligibility Criteria

The eligibility criteria for the systematic review and meta-analysis follows the PICO element [15,16]:-P: Pregnant healthy woman;-I: Exercise (aerobic exercise, resistance exercise, aerobic and resistance combined, or mind–body exercise);-C: Usual care or Exercise;-O: Gestational diabetes mellites (GDM), preeclampsia and spontaneous abortion, withdrawals from the study, adverse events (all adverse events).

The population was healthy pregnant women with no current diseases at study enrollment. Studies including participants with chronic or acute medical conditions, hypertension, preeclampsia, diabetes type 1, type 2, or GDM at study enrollment were excluded. Studies including participants with GDM in earlier pregnancies were allowed, if the woman was healthy at study enrollment.

The intervention was exercise during pregnancy. Exercise was defined by ACSM in the *Guidelines for Exercise Testing and Prescription* as “a type of physical activity consisting of planned, structured, and repetitive bodily movement done to improve and/or maintain one or more components of physical fitness” [17]. In this systematic review, exercise interventions were divided into four subcategories, classified as aerobic exercise, resistance exercise, aerobic and resistance combined, or mind–body exercise.

○Aerobic exercises include interventions such as swimming, cycling, jogging, dancing, and running.○Resistance exercises include interventions aiming at increasing muscle strength by progressive use of loads (e.g., free weights, elastic bands, weight machines, or body weight).○Combined exercise intervention contains both aerobic and resistance exercises.○Mind–body exercise interventions include activities integrating low-intensity muscular activity (e.g., flexibility, balance, coordination, yoga, Pilates, tai chi, and qigong).

Any study setting or context was eligible.

The comparators were no exercise, usual care, advice to stay active, or other active comparative interventions (as listed above). Studies with co-interventions such a dietary intervention/advice were included.

The primary outcomes were GDM, preeclampsia, and spontaneous abortion at the end of the intervention or at birth if the intervention continued postpartum.

GDM was defined as any level of glucose intolerance with inception or first recognition during pregnancy [18]. Preeclampsia was defined by WHO as a new onset of hypertension during pregnancy, characterized by persistent hypertension (diastolic blood pressure ≥90 mm Hg) and substantial proteinuria (>0.3 g/24 h), often occurring after gestational week 20 [19]. Spontaneous abortion was defined by pregnancy loss before twenty weeks of gestation (often also referred to as miscarriage) [20]. There was no restriction regarding the methods to diagnose GDM, preeclampsia, and spontaneous abortion in the included studies. Table 1 shows the individual studies and their methods of diagnosing their outcomes. Secondary outcomes were withdrawals from the study and adverse events, and all types of adverse events were included in the study.

Eligible study designs were randomized controlled trials (RCT), randomized cross-over trials, and cluster randomized trials comparing different types of exercise interventions or comparing exercise interventions to control treatment. There were no restrictions regarding year of publication or language in the search string, and articles in other languages than Danish, English, Swedish, and Norwegian were unsorted in title abstracts [16].

### 2.3. Search Strategy

The electronic systematic literature search was conducted from the 22nd to the 23rd of February 2022 with no restrictions on the initiation date. The following databases were searched: MEDLINE (via PubMed), EMBASE (via Ovid), Cochrane Central Register of Controlled Trials (CENTRAL), and SPORTDiscus, to identify eligible RCT studies.

As a second step of the literature search, Web of Science (via Web of Knowledge) was searched using citation tracking, and reference lists from selected articles and systematic reviews were checked for additional relevant articles. The search terms were included as words in the title, abstract, and Medical Subject Heading (MeSH). Study authors were not contacted to identify additional studies.

### 2.4. Study Selection

Titles and abstracts of identified studies were screened, followed by an evaluation of relevant full-text articles in accordance with the pre-specified criteria. The screening of literature was conducted by three reviewers (C.P.P., E.B., H.E.C). Each study was reviewed by two authors independently. Any disagreement was solved by discussion or counseling with C.J. or M.N.H. All citations obtained from the search strategy were imported into EndNote and duplicates were removed. Covidence software was used for managing and selection of references identified from the databases.

### 2.5. Data Extraction

For data collection, a customized Microsoft Excel Sheet was made, and all data were independently extracted by two out of three authors (C.P., H.E.C. and E.B.). Any disagreement between authors was solved by a fourth author (C.J. or M.N.H.). The study information was used in the way the study presented the data items. No assumptions nor simplifications of data were made, and study authors were not contacted to confirm data. If multiple publications of a single study were identified, the article with the most complete data was included in the meta-analysis. Data extraction was performed on source (review’s author and year of publication), method (design, published year, and study setting), participants (total number of participants as well as the number of participants allocated to the intervention and control groups), demographic characteristics (age (mean) and body mass index (BMI) (mean)), interventions (duration of the treatment and details about how the treatments were performed in the intervention and control group(s)), modality of the intervention (aerobic, resistance, combined aerobic and resistance, or body–mind), intensity (comprehensive extensive, moderate extensive, and less extensive (Comprehensive extensive >24 sessions; ≥60 min/session; ≥2×/week; Aerobic exercise (AE): ≥70% of heart rate maximum (HRM), the maximal oxygen uptake (VO_2_ max) heart rate reserve (HRR); resistance exercise (RE): ≥70% of 1 repetition maximum RM. Moderate extensive: 12–24 sessions; 30–60 min/session; 1–2×/week; AE: 50–70% of HRM, VO_2_max, or HRR; RE: 50–70% of 1 RM. Less extensive: <12 sessions; ≤1×/week; ≤30 min/session; AE: ≤50% of HRM, VO_2_ max, HRR; RE: ≤50% of 1 RM), amount of supervision (supervised, non-supervised, or mixed interventions), and number of pregnant women with outcomes.

### 2.6. Risk of Bias Assessment

The overall risk of bias was assessed independently by two out of three authors (C.P., E.B., H.E.C.) using the Cochrane Risk of Bias tool 2.0 (RoB 2.0 [21]. Discrepancies were resolved through discussion or counseling with C.J. or M.N.H.

The Risk of Bias crib sheet for individually randomized, parallel-group trials was used (version from August 2019), analyzing the ‘intention-to-treat’ effect. Five domains were assessed: the randomization process, deviations from the intended interventions, missing outcome data, measurement of the outcome, and selection of the reported result [21]. The overall risk of bias was rated as high if one of the domains was associated with a high risk of bias, or if a minimum of four out of five of the domains were associated with some concerns.

### 2.7. Effect Measures

The effect sizes for all outcomes were expressed by relative risk (RR) and 95% confidence interval of the interventions.

### 2.8. Synthesis Methods

The meta-analysis was performed with a standard pairwise using a random effect model meta-analysis (REML) for direct comparisons between interventions and control group, due to an expected heterogeneity between studies because of differences in interventions, settings, and participants’ characteristics. Forest plots for all outcomes were made to illustrate group-specific results. In cases of studies with multiple interventions, the interventions were analyzed separately in the analysis. Funnel plots were used to evaluate and illustrate the risk of publication bias (i.e., small study bias), and the Egger’s test was performed to test potential small study bias [14].

Three subgroup analyses were conducted (intensity of the intervention, intervention modalities, and supervised/non-supervised/mixed interventions (combination of supervised and non-supervised interventions)).

To explore the heterogeneity, three pre-specified meta-regression analyses (intervention length, body mass index (BMI), and age) and four post hoc sensitivity analyses (training sessions per week, length of trainings sessions (min), number of supervised sessions, and end of intervention (gestational week) were conducted for GDM and preeclampsia.

The between-study variance (τ^2^) was estimated, and a decrease in τ^2^ of 10% or more was considered to have a relevant impact on the outcome in reducing the heterogeneity. Bubble plots were made for all significant variables to display a visual overview of the distribution of observed effects [14].

Inconsistency was assessed using I^2^, describing the percentage of total variation across the studies due to heterogeneity rather than chance (low inconsistency: I^2^ < 40%, moderate inconsistency: I^2^ between 30 and 60%, substantial inconsistency: I^2^ between 50 and 90%, large inconsistency: I^2^ > 75–100%). The analyses were conducted in STATA 17.0.

### 2.9. Certainty Assessment

The Grading Recommendations Assessment, Development, and Evaluation (GRADE) guidelines were used to evaluate the overall certainty of evidence, with four possible ratings: high, moderate, low, and very low. RCTs start at high certainty of the effect estimates. Downgrading was performed in cases of the following: study limitation, inconsistency, indirectness, imprecision, and/or publication bias [21].

## 3. Results

### 3.1. Study Selection

The initial search yielded 4490 studies. After removing duplicates, 2984 studies were screened for title/abstracts. A total of 187 studies were identified as potentially eligible and were screened in full text resulting in 20 studies included in this systematic review (Figure 1). A list of excluded studies with reasons given is provided in the Appendix A.

### 3.2. Study Characteristics

The 20 identified studies included a total of 6767 participants [12,22,23,24,25,26,27,28,29,30,31,32,33,34,35,36,37,38,39,40]. The studies were published between 2012 and 2021, in ten different countries (Ireland, India, Spain, Brazil, Canada, New Zealand, USA, Norway, Australia, Croatia). The main characteristics of the 20 included studies are presented in Table 1.

All studies compared the intervention to a control group with no active intervention. The duration of the interventions varied from 16 weeks to 31.5 weeks. Most of the interventions lasted until birth or close to (gestational week 38). Eight studies were conducted at a hospital [22,23,24,25,26,27,28,29] and four studies were conducted in health care centers [30,31,32,33]. The remaining studies took place either at home, outside, or in a gym [34,35,36,37,38,39] and two studies did not report settings [12,40]. Fifteen studies had three training sessions per week [12,22,23,24,25,26,27,28,29,30,32,34,35,38,40] whereas the remaining five had two [31], four [33,36,37], or seven [39] sessions per week. No studies reported comprehensive intensity. The intensity of the exercise was moderate in all studies except two where the intensity was less extensive [28,39]. Seventeen interventions were supervised [12,22,23,24,25,26,27,28,29,30,31,32,33,34,35,38,40] and in three studies, the sessions were mixed [36,37,39], and none were unsupervised. The interventions comprised of aerobic [22,24,35,36,38], aerobic and resistance [12,23,25,26,27,29,30,31,32,33,34,37,40], and mind–body [28,39]. No studies reported resistance training alone.

GDM was obtain by medical records in six studies [12,25,27,30,35,40], and in ten studies, the authors obtained the data themselves according to WHO definitions of GDM [22,24,26,28,29,31,32,33,36,37]; one study did not report how the data were obtained [23], and in one study, the participants reported the outcome themselves [34].

In six of the studies, preeclampsia was obtained from medical records [22,24,25,28,31,36,39]; in one study, it was self-reported [34], and one study did not report how they obtained the data [34]. Two studies included women with GDM in earlier pregnancies but only if they were healthy at baseline [12,24]. None of the studies had any dietary interventions; only advice or a pamphlet on healthy eating was given to both the intervention and control group.

**Table 1 ijerph-20-06069-t001:** Study characteristics of the included studies.

Author (Year)	Setting and Samples	Intervention Group	Control Group	Outcome	Conflict of Interest
Daly (2017) [29]	n: 88 Age (mean): 29.7 BMI: 34.7 Duration of intervention (weeks): 21 Setting: Hospital Country: Ireland	Three medically supervised exercise classes per week for the duration of their pregnancy. Women received an invitation to a Facebook group to create a community among participants. The program consisted of 50–60 min of exercise with a 10 min warm-up, 15–20 min of resistance or weights, 15–20 min of aerobic exercises, and a 10 min cool-down.	The control group received standard hospital-written information on exercise. As part of routine prenatal care in Ireland, all women received a pamphlet with information on healthy eating based on national guidelines for nutrition and pregnancy.	GDM: A standard 75 g oral glucose tolerance test (OGTT). Based on the International Association of Diabetes in Pregnancy Studies Group criteria.	The authors report no conflict of interest.
Rakhshani (2012) [28]	n: 93 Age (mean): 27.5 BMI: 25.22 Duration of intervention (weeks): 16 Setting: Hospital Country: India	The intervention group received standard care plus one-hour yoga sessions, three times per week.	The control group received standard care plus walking for half an hour, mornings and evenings (the routine antenatal exercise offered by the hospital).	GDM: New onset of glucose intolerance during pregnancy (serum glucose ≥5.5 mmol/L). Preeclampsia: Pregnancy-induced hypertension with new onset of proteinuria (24 h urine protein ≥300 mg).	The authors report no conflict of interest.
Barakat (2012) [32]	n: 100 Age (mean): 31.5 BMI: 22.85 Duration of intervention (weeks): 31.5 Setting: Health care center Country: Spain	The intervention group exercised 35–45 min, three times per week, with two land aerobic sessions and one aquatic activities session.	The control group did not perform any type of exercise, except activities needed for daily living.	GDM: Non-fasting oral glucose challenge test was used to screen, in which venous blood was sampled 1 h after a 50 g oral glucose load. If the 1 h glucose result was at least 7.8 mmol/L, the participant was referred for a 100 g fasting glucose 3 h tolerance test. Normal results were a blood glucose below 5.3 mmol/L at baseline, below 10 mmol/L at 1 h, below 8.6 mmol/L at 2 h, and below 7.8 mmol/L at 3 h.	The authors report no conflict of interest.
DeOliveriaMelo (2012) [38]	Group A n: 93 Age (mean): 24 BMI: 24.15 Duration of intervention (weeks): 25 Setting: Outside Country: Brazil Group B n: 94 Age (mean): 25 BMI: 23.45 Duration of intervention (weeks): 18 Setting: Outside Country: Brazil	The intervention group received a supervised intervention three times per week. The initial duration of walking was 15 min, progressively increasing over the study period in accordance with the woman’s previous physical fitness level. Intensity of the exercise was indeed moderate: heart rate between 60% and 80% of maximum heart rate, corrected for age.	The control group continued to receive prenatal care at their health care unit.	Preeclampsia: Not reported.	The authors report no conflict of interest.
Pais (2021) [39]	n: 132 Age (mean): 27.85 BMI: Duration of intervention (weeks): 20 Setting: Home Country: India	The interventioin group had a yoga session of 45 min daily at home until birth. Follow-ups were weekly by telephone. The women were instructed to maintain a yoga therapy diary and a pranayama log sheet.	The control group received routine standard care.	Preeclampsia: Was defined as systolic blood pressure, higher than 140 mm Hg or diastolic blood pressure higher than 90 mm Hg; readings recorded in resting (i.e., sleeping position 24 h apart).	The authors report no conflict of interest.
Hui (2012) [37]	n: 224 Age (mean): 29.4 BMI: 25.3 Duration of intervention (weeks): 16 Setting: Community center/Home Country: Canada	Community-based group exercise sessions (floor aerobics, stretching and strength exercises), instructed home exercise (an exercise instruction video created for pregnant women was provided to assist home exercise), and dietary advising.	Participants in the control group received usual prenatal care and were provided with a packaghe of up-to-date information on physical activity and healthy diet.	GDM: According to the 2008 Guidelines of the Canadian Diabetes Association [1].	The authors report no conflict of interest.
Pelaez (2019) [27]	n: 345 Age (mean): 31.31 BMI: 23.8 Duration of intervention (weeks): 24 Setting: Hospital Country: Spain	Structured, supervised exercise program 3 times per week, with a duration of 60 to 65 min. Each session started with an 8 min warm-up that consisted of walking, movement games, light dancing-based warm-up, and gentle dynamic stretching. The core section of the training session lasted 35 min and included low-impact aerobics and 10 min of major muscle group resistance training.	The control group received usual care, (follow-ups by midwives, advice about nutrition, and physical activity from health care professionals. Women in this group were not prevented from exercising on their own.	GDM: Obtained from perinatal obstetric records.	The authors report no conflict of interest.
Seneviratne (2016) [36]	n: 75 Age (mean): 31.35 BMI: 33.1 Duration of intervention (weeks): 16 Setting: Home Country: New Zealand	The intervention group participated three to five times per week in a structured home-based moderate-intensity antenatal exercise program utilizing magnetic stationary bicycles between 15 and 30 min per session.	Routine antenatal and delivery care with their chosen maternity carers. Participants in both groups were able to continue their routine physical activity and diet without restriction.	GDM: Gestational diabetes mellitus was diagnosed by a standard 75 g oral glucose tolerance, based on a fasting plasma glucose ≥5.5 mmol/L and/or a 2 h post test plasma glucose ≥9.0 mmol/L. Preeclampsia: Pregnancy-induced hypertension with proteinuria.	The authors report no conflict of interest.
Mcdonald (2021) [35]	n: 128 Age (mean): 30.25 BMI: 25.7 Duration of intervention (weeks): 20 Setting: University-affiliated gyms Country: USA	The intervention group participated in individual, supervised, moderate-intensity (40–59% of maximal oxygen uptake) aerobic exercise sessions for 50 min, three times per week.	The control group was offered the opportunity to attend low- intensity stretching classes.	GDM: Electronic health records.	The authors report no conflict of interest.
Price (2012) [33]	n: 91 Age (mean): 29.05 BMI: 27.65 Duration of intervention (weeks): 23 Setting: St. David’s Cardiac Rehabilitation and Fitness Center Country: USA	The intervention contained a program of supervised aerobic exercise of 45–60 min duration, performed four times per week at moderate intensity.	The control group only did work or household chores. They were told not to exercise because it would blur the distinction between groups. All subjects were told to follow the dietary advice of their obstetricians or midwives, and there was no attempt to estimate calorie intake.	GDM: Two or more abnormal values on a 3 h, 100 g glucose tolerance test.	The authors report no conflict of interest.
Sagedal (2017) [31]	n: 606 Age (mean): 28 BMI: 23.65 Duration of intervention (weeks): 21 Setting: Health care clinics Country: Norway	Dietary counseling twice by telephone and access to twice weekly exercise groups. Exercises started with 10 min of warm-up, 40 min of strength training and cardiovascular exercise at moderate intensity (using aerobics, calisthenics, and weight training), and 10 min of stretching.	The control group received routine prenatal care in accordance with Norwegian standards. All participants received a booklet with advice on prenatal nutrition and physical activity, including recommendations for weight gain.	GDM: A glucose tolerance test was performed measuring serum glucose after fasting and at 2 h after an intake of 75 g of glucose. Glucose levels ≥7.8 mmol/L at 2 h were classified as elevated, based on both national and WHO criteria. Preeclampsia: Data based on complete medical record review.	The authors report no conflict of interest.
Barakat (2019) [30]	n: 520 Age (mean): 31.4 BMI: 23.58 Duration of intervention (weeks): 30 Setting: Health care center Country: Spain	The intervention group received usual care and a structured and supervised moderate exercise intervention program three times per week (55–60 min per session). The main section was 30–35 min in length and included moderate-intensity aerobic exercises and resistance exercises.	The control group received usual obstetric care from health professionals.	GDM: Collected from hospital records.	The authors report no conflict of interest.
Barakat (2013) [26]	n: 510 Age (mean): 31.3 BMI: 23.9 Duration of intervention (weeks): 27 Setting: Hospital Country: Spain	The intervention group exercised 3 days per week, 50–55 min per session. The intervention involved aerobic exercises, muscle strength, and flexibility, and met the standards of the American College of Obstetricians and Gynecologists.	The control group received general advice from their midwife about the positive effects of physical activity. The control group received usual care, the same as in the exercise group. Women were not prevented from exercising on their own.	GDM: GDM diabetes was diagnosed according to the two accepted criteria: - The WHO criteria, that is, 2 h glucose ≥7.8 mmol/L; - The International Association for Diabetes in Pregnancy Study Group (IADPSG), that is, 2 h glucose ≥8.5 mmol/L.	The authors report no conflict of interest.
Barakat (2016) [25]	n: 840 Age (mean): 31.7 BMI: 23.5 Duration of intervention (weeks): 28 Setting: Hospital Country: Spain	The intervention group exercises three days per week (50–55 min per session). The intervention involved aerobic exercise, aerobic dance, muscular strength, and flexibility, and met the standards of the American Congress of Obstetricians and Gynecologists.	The control group received usual care with health care providers during pregnancy, which was equal to the intervention group. Women were not prevented from exercising on their own.	GDM: Medical records. Preeclampsia: Medical records.	The authors report no conflict of interest.
Cordero (2015) [12]	n: 342 Age (mean): 33.25 BMI: 23.0 Duration of intervention (weeks): 26 Setting: Not reported Country: Spain	The intervention group exercised for 50 to 60 min sessions, three times per week, two times on land (gym hall) and one time as an aquatic water-based activity.	Received usual care and remained inactive.	GDM: Medical records.	The authors report no conflict of interest.
Wang (2017) [24]	n: 300 Age (mean): 32.32 BMI: 26.8 Duration of intervention (weeks): 27 Setting: Hospital Country: Australia	The intervention group engaged in a supervised moderate-intensity cycling program at least three times per week.	The control group continued with usual care, and were not prevented from participating in exercise sessions on their own. All women received standard prenatal care throughout the intervention period, and they had equal numbers of usual visits with their obstetricians during pregnancy.	GDM: 75 g oral glucose tolerance test (OGTT) after an overnight fast. GDM was diagnosed when any value was ≥5.1 mmol/L at start, ≥10.0 mmol/L at 1 h, or ≥8.5 mmol/L at 2 h. Values of 7.0 mmol/L at 0 h or 11.1 mmol/L at 2 h were diagnosed as diabetes mellitus, regardless of pregnancy stage Preeclampsia: Defined as new- onset hypertension (systolic blood pressure higher than 140 mm Hg or diastolic blood pressure higher than 90 mm Hg) and new-onset proteinuria (300 mg of protein in 24 h or a urine protein/creatinine ratio of 0.3 mg/dL) > 20 weeks gestation or, in the absence of proteinuria, new-onset hypertension with new-onset thrombocytopenia, renal insufficiency, impaired liver function, pulmonary edema, or cerebral or visual disturbances).	The authors report no conflict of interest.
Barakat (2014) [23]	n: 251 Age (mean): 31.54 BMI: 23.9 Duration of intervention (weeks): 28 Setting: Hospital Country: Spain	The intervention group received a supervised aerobic activity physical conditioning program that included three 55 to 60 min sessions per week with light- to moderate-intensity. Each session consists of 25–30 min of cardiovascular exercise, 10 min of specific exercises (strength and balance exercises), and 10 min of pelvic floor muscle training.	The control group did not exercise during this period; they received the usual information provided by their midwives or health care professionals.	GDM: Not reported.	Not reported.
GinardaSilva (2017) [34]	n: 639 Age (mean): 27.15 BMI: 25.15 Duration of intervention (weeks): 16 Setting: Gym of the physical education school Country: Brazil	Women in the intervention group received a structured, individually supervised, moderate-intensity exercise program for 1 h, 3 days per week. Each session involved warm-up, aerobic activities, strength training, and stretching exercises.	The control group received standard antenatal care and were encouraged to continue their normal daily activities.	GDM: Self-reported and evaluated during the hospital stay at delivery Preeclampsia: Self-reported.	The authors report no conflict of interest.
Ruiz (2013) [40]	n: 962 Age (mean): 31.75 BMI: 23.6 Duration of intervention (weeks): 30 Setting: Not reported. Country: Spain	The exercise intervention group trained 3 days per week, 50–55 min per session. The intervention involved light to moderate aerobic exercises, muscle strength and flexibility, and met the standards of the American College of Obstetricians and Gynecologists.	Participants in the control group had their usual visits with health care providers during pregnancy. Women were not discouraged from exercising on their own.	GDM: Medical records.	Not reported.
Tomic (2013) [22]	n: 334 Age (mean): 29.1 BMI: 22.9 Duration of intervention (weeks): 30 Setting: Hospital Country: Croatia	Exercise was performed 3 times per week and was regular aerobic exercise that consisted of a warm-up period (5 min), aerobic exercise (30 min), stretching (10 min), and a cool-down period (5 min).	The control group did not participate in any organized regular physical exercise during the trial.	GDM: A 50 g glucose test. Blood glucose was measured one hour after glucose intake and the value of >7.8 mmol/L was considered significant, and diabetes was confirmed by a glucose tolerance test. Preeclampsia: Defined as persistently elevated blood pressure (diastolic blood pressure of 90 mm Hg or higher and systolic pressure of 140 mm Hg and higher on more than two occasions) with proteinuria or edema or both.	The authors report no conflict of interest.

### 3.3. Completeness of Data

Eighteen studies reported the outcome of GDM [12,22,23,24,25,26,27,28,29,30,31,32,33,34,35,36,37,40] and nine studies reported the outcome of preeclampsia [22,24,25,28,31,34,36,38,39]. No studies reported the outcome of spontaneous abortion. All twenty studies reported the outcome of withdrawal, and only eight studies reported adverse events [23,24,27,30,32,34,36,37].

All data were obtained for the covariates and only one study failed to report BMI [39].

### 3.4. Risk of Bias in Studies

The overall risk of bias assessment is presented in Table 2. Fifteen studies were judged with some concern and five with high risk of bias. Nine studies did not pre-specify the outcome in the protocol, including how they would measure it, and therefore, the studies were rated as having some concerns in domain 5 (bias in selection of the reported result) [22,25,26,28,29,33,35,37,39]. None of the included studies were able to blind the participants to the allocation of groups due to the type of intervention applied (exercises). The outcomes of preeclampsia and GDM were checked for small study bias with funnel plots (see Appendix B). The visual assessment of the funnel plots indicates a risk of small study bias, especially in GDM. Egger’s tests found no significant small study bias for preeclampsia (*p* = 0.16) or GDM (*p* = 0.08).

### 3.5. Results of Synthesis

#### 3.5.1. Gestational Diabetes Mellitus

Eighteen studies reported the effect of exercise intervention versus control group on GDM. The crude meta-analysis showed a significant effect favoring exercise, relative risk: 0.66 (95% CI: 0.50 to 0.86) with moderate heterogeneity (I^2^ = 47.68%) (Figure 2). No subgroup differences were seen regarding modality (*p* = 0.21), intensity (*p* = 0.20), or supervision/non-supervision/mixed (*p* = 0.38) (Appendix C).

To explore the heterogeneity a meta-regression was conducted on age, BMI, duration of intervention (weeks), training sessions per week, length of training sessions (min), number of supervised sessions, and end of intervention (gestational week) (Table 3 and Figure 3). A beneficial effect of exercise on risk reduction in GDM were shown among older women, women with lower BMI, women who received a longer duration of intervention, and a higher number of supervised training sessions (Table 3, Figure 3).

#### 3.5.2. Preeclampsia

Nine studies reported the effect of exercise intervention versus control group on preeclampsia. The crude meta-analysis showed no effect of exercises on preeclampsia (RR 0.65, 95% CI: 0.42 to 1.03) with low heterogeneity (I^2^ = 18.75%) (Figure 4).

When investigating intervention modality, the largest effect was seen following body–mind intervention (RR = 0.16, 95% CI: to 0.04 to 0.58), whereas no effect was seen following a combination of aerobic and resistance intervention (RR = 0.60, 95% CI: 0.27 to 1.35), aerobic intervention alone (RR = 1.01, 95% CI: 0.52 to 1.96). None of the included studies reported on resistance modality alone. No significant subgroup differences were seen between intervention with supervision and a combination of non-supervised and supervised intervention (*p* = 0.21) (Appendix D).

Studies using low intensity seemed to be more effective in risk reduction of preeclampsia compared to studies using moderate intensity (*p* = 0.02) (Appendix D).

The variables of mean age, duration of intervention, length of training intervention, and number of supervised sessions showed an increase of tau^2^ meaning that these variables could not explain the heterogeneity (Table 4).

#### 3.5.3. Withdrawals

Twenty studies reported withdrawals from the exercise interventions and control group. The crude meta-analysis showed no differences in withdrawals between the exercise and control group (RR 0.97, 95% CI: 0.84 to 1.10) (Figure 5).

#### 3.5.4. Adverse Events

Eight studies investigated adverse events, but no adverse events were reported in any of the studies [23,24,27,30,32,34,36,37]. In five studies, they looked at adverse events on maternal and fetal wellbeing; one study looked at major adverse events, and the last two looked at all adverse events.

#### 3.5.5. Certainty of Evidence

Table 5 presents the assessments of the certainty in the body of evidence for each outcome assessed. The overall certainty of the evidence for the outcome GDM was moderate because of high probability of publication bias. The certainty was high for withdrawal. The certainty was low for preeclampsia, as the confidence intervals include both appreciable benefit and appreciable harm.

## 4. Discussion

Overall, our meta-analyses including more than 5700 participants showed that exercise during pregnancy is beneficial in the prevention of GDM. In the prevention of GDM, all included modalities (aerobic, resistance, aerobic and resistance combined, or mind–body exercise), delivery mode (supervised or mixed), and intensities (moderate extensive (12–24 sessions; 30–60 min/session; 1–2×/week; AE: 50–70% of HRM, VO_2_max or HRR; RE: 50–70% of 1 RM) and less extensive (<12 sessions; ≤1×/week; ≤30 min/session)), seem equally effective. In the prevention of preeclampsia, subgroup analysis supports an association between mind–body exercise and exercise at low intensity with reduced risk of preeclampsia. There were no concerns about harms; no studies reported spontaneous abortion as an outcome and rare adverse events may not occur due to the low number of participants included in the studies. The overall certainty of the evidence for the outcome of GDM was moderate because of a high probability of publication bias, and low for preeclampsia, due to the confidence intervals which includes both appreciable benefit and appreciable harm. The certainty of evidence for withdrawal was high.

### 4.1. Comparison to Other Systematic Reviews and Clinical Guidelines

None of the previously published systematic reviews on the effect of exercise on GDM and preeclampsia have analyzed the effect of exercise modality. Five systematic reviews have previously investigated the association between performing exercises and the risk of developing GDM (Zheng et al., 2007 [41], Ming et al., 2018 [42], Yu et al., 2018 [43], Nasiri-Amiri et al., 2019 [44], and Yin et al., 2013 [45]). Three out of five systematic reviews showed a significant reduction in the risk of GDM [41,42,43], which supports the finding in our review. Still, most of the reviews included the same low number of studies. The risk reduction of GDM due to exercise was higher in most of the systematic reviews compared to ours. Four of the reviews [41,42,43,45] included only normal-weight participants, where some had GDM at baseline, which could explain the larger risk reduction observed in these studies. The review by Nasiri-Amiri et al. was the one with the most included studies (eight studies). This review only included obese and overweight pregnant women, and found a lower reduction in risk of GDM for the exercise interventions compared to our study. This shows that obese or overweight women may not receive the same effect of exercise intervention as normal-weight women. This is also in alignment with the meta-regression in our study showing that an increase in BMI decreased the effect of the exercise intervention. One of the main limitations in all previous systematic reviews was the low number of included studies (between 4 and 8 studies), whereas our study included 18 studies investigating the outcome of GDM.

Four systematic reviews investigated the association between exercise and the risk of developing preeclampsia. A Cochrane review by Kramer et al. [46] from 2006 showed no effect of performing aerobic exercise in the risk of preeclampsia for healthy pregnant women. Only one study was included in the meta-analysis, and therefore, there is a high risk of imprecision and it must be interpreted with large caution [46]. The Cochrane review by Meher et al., 2006 [47] also showed no effect of exercise on the risk of preeclampsia. Meher et al. included participants with preeclampsia in earlier pregnancy. These women have a higher risk of getting preeclampsia during their next pregnancy and may therefore receive potential benefit from exercise. Similar to our review, the results from Kasewara et al. indicate that physical activity could be used to prevent preeclampsia; however, the included studies were all cohort and case control, and most of the data were collected through a questionnaire postpartum, which increases the risk of recall bias [48]. The last systematic review by Syngelaki et al., 2018 [49] included only obese or overweight participants (three studies included), and this review showed no clear differences between the intervention and control group on preeclampsia.

### 4.2. Strengths and Limitations

The systematic standardized method and risk of bias assessment of the included studies is one of the strengths of our review. The review follows the guidelines from the *Cochrane Handbook for Systematic Review of Interventions* and is reported according to the PRISMA guidelines [13,14]. Additionally, the review was protocolized at PROSPERO prior to the search. A comprehensive literature search was performed, and the study selection, risk of bias assessment, and data extraction were performed independently by two authors.

Four studies were excluded due to language, however, the expected impact on the results is considered minimal. In addition, the prespecified network meta-regression analyses were not carried out due to a lack of studies comparing different exercise interventions. Besides that, it was not protocolized when a decrease of τ^2^ was sufficient, at what scale to extract the outcomes, or how to handle missing data.

Future studies should focus on intervention with resistance exercise alone, non-supervised exercise, and comprehensive extensive exercise, due to a lack of knowledge about how these affect the outcomes of GDM, preeclampsia, and spontaneous abortion. The lack of studies about these topics also influences the generalizability of our study, meaning that we cannot say how, for example, resistance exercises alone affect the outcomes compared to aerobic; knowledge like this would be useful for pregnant women and therapists when planning an exercise program.

## 5. Conclusions

Overall, our meta-analyses showed that exercise consisting of aerobic, aerobic and resistance combined, or mind–body exercise during pregnancy are equally beneficial and safe in the prevention of GDM. Moderate and less intense interventions and supervised or mixed delivery mode seem equally effective in prevention of GDM.

In the prevention of preeclampsia, subgroup analyses support an association between mind–body exercise and exercise at low intensity and low risk of preeclampsia, but more high-quality randomized studies are needed.

## Figures and Tables

**Figure 1 ijerph-20-06069-f001:**
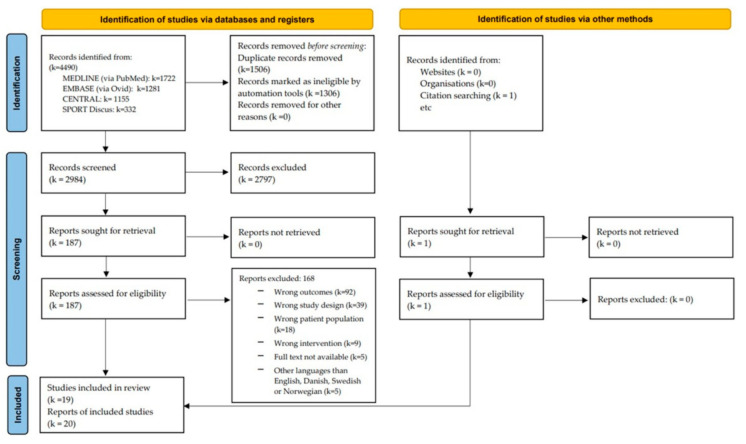
Prisma flowchart over study selection [13].

**Figure 2 ijerph-20-06069-f002:**
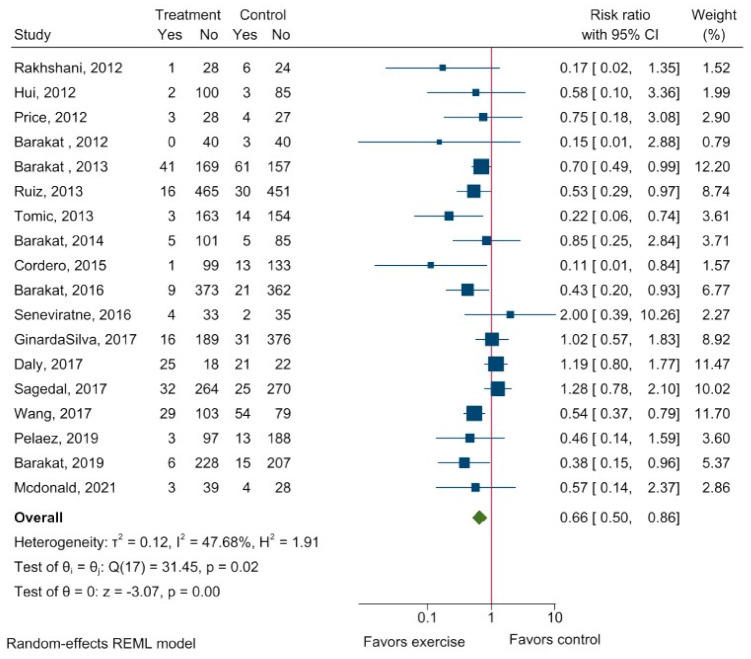
Risk of GDM after exercise intervention during pregnancy. Explanation: 95% CI−95% confidence interval. The forest plot shows the crude risk of GDM after exercise. Relative risk <1 favors exercise, whereas relative risk > 1 favors control intervention. I^2−^inconsistency; τ^2^−between-study variance. Treatment−intervention group. Yes—refers to all cases with GDM; no−refers to the cases without GDM [12,22,23,24,25,26,27,28,29,30,31,32,33,34,35,36,37,38].

**Figure 3 ijerph-20-06069-f003:**
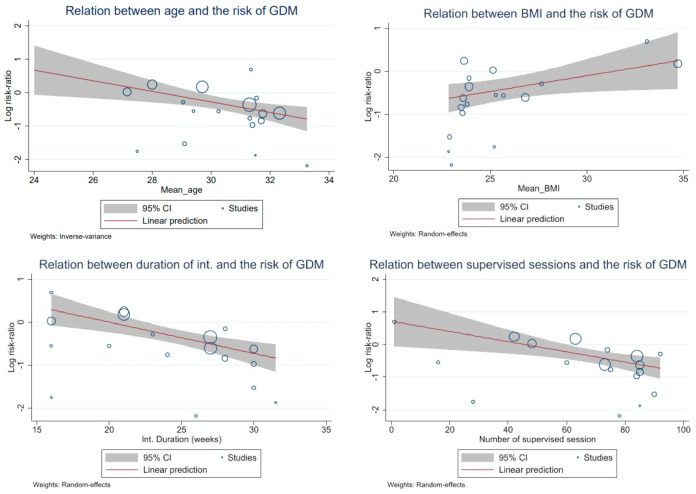
Relation between covariates and the risk of GDM. Explanation: 95% CI−95% confidence interval. The size of the bubbles represents the precision of the studies, the bigger the bubble, the higher the precision. GDM—gestational diabetes mellitus.

**Figure 4 ijerph-20-06069-f004:**
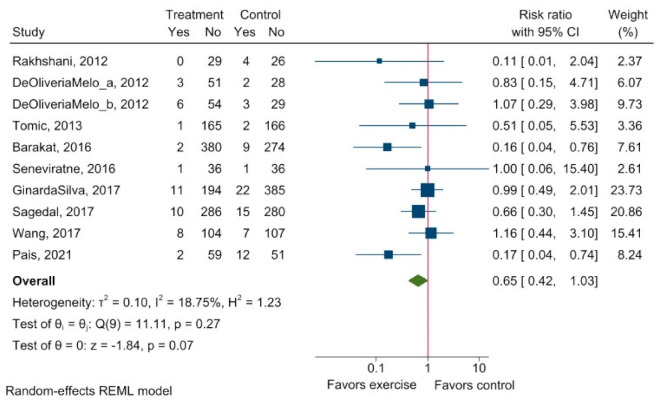
Risk of preeclampsia after exercise intervention during pregnancy. Explanation: 95% CI—95% confidence interval. The forest plot shows the crude risk of preeclampsia after exercise. Relative risk <1 favors exercise, whereas relative risk >1 favors control intervention. I^2^−inconsistency; τ^2^−between-study variance. Treatment−intervention group. Yes−refers to all cases with preeclampsia; no−refers to the cases without preeclampsia [22,24,25,28,31,34,36,38,39].

**Figure 5 ijerph-20-06069-f005:**
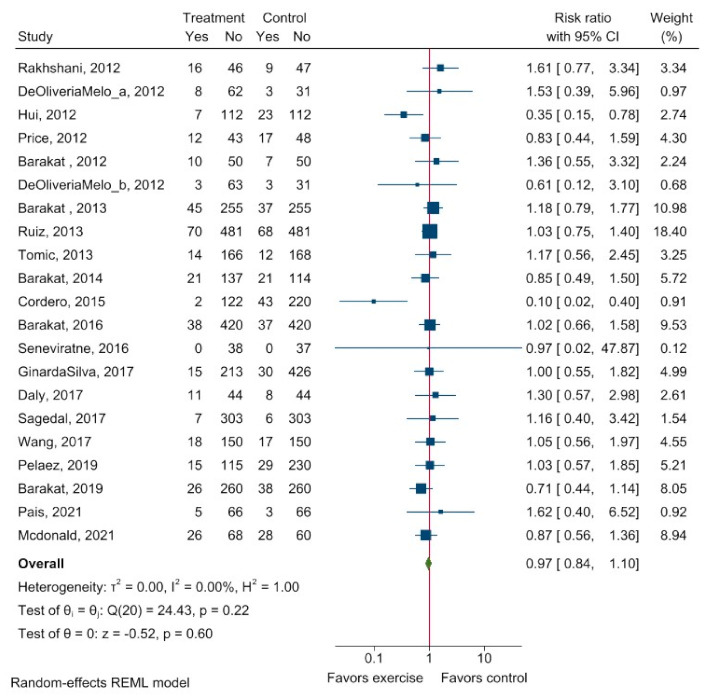
Risk of withdrawal after exercise intervention compared to control group during pregnancy. Explanation: 95% CI−95% confidence interval. The forest plot shows the crude risk of withdrawal after exercise. Relative risk <1 favors exercise, whereas relative risk >1 favors control intervention. I^2^—inconsistency; τ^2^—between-study variance. Treatment—intervention group. Yes—refers to all cases with withdrawal; no—refers to the cases without withdrawal [12,22,23,24,25,26,27,28,29,30,31,32,33,34,35,36,37,38,39,40].

**Table 2 ijerph-20-06069-t002:** Risk of bias evaluation based on Cochrane Risk of Bias tool, version 2 (ROB 222) [12,22,23,24,25,26,27,28,29,30,31,32,33,34,35,36,37,38,39,40].

Study	D1	D2	D3	D4	D5	Overall
Daly et al., 2017 [29]						
Rakhshani et al., 2012 [28]			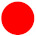			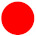
Barakat et al., 2012 [32]						
DeOliveriaMelo et al., 2012 [38]						
Pais et al., 2021 [39]				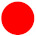		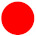
Hui et al., 2012 [37]						
Pelaez et al., 2019 [27]						
Seneviratne et al., 2016 [36]						
Mcdonald et al., 2021 [35]						
Price et al., 2012 [33]						
Sagedal et al., 2017 [31]						
Barakat et al., 2019 [30]						
Barakat et al., 2013 [26]						
Barakat et al., 2016 [25]						
Cordero et al., 2015 [12]						
Wang et al., 2017 [24]						
Barakat et al., 2014 [23]						
GinardaSilva et al., 2017 [34]				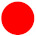		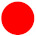
Ruiz et al., 2013 [40]			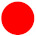			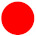
Tomic et al., 2013 [22]	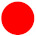					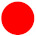

Domains: D1: Randomisation process; D2: Deviations from the intended interventions; D3: Missing outcome data; D4: Measurement of the outcome; D5: Selection of the reported result; Judgement: Low: 

 Some concerns: 

 High: 
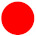
.

**Table 3 ijerph-20-06069-t003:** Evaluation of covariates for the effect of exercise during pregnancy on GDM.

	k	OR (95% CI)	*p*-Value	τ^2^	I^2^
Overall effect	18	0.60 (0.45 to 0.81)		0.1476	42.26%
Mean age	18	0.83 (0.74 to 0.93)	0.001	1.9 × 10^−7^	0%
Mean BMI	18	1.10 (1.00 to 1.21)	0.043	0.1434	41.22%
Duration of intervention (weeks)	18	0.93 (0.89 to 0.97)	0.001	0.007592	3.48%
Training sessions per week	18	0.75 (0.38 to 1.48)	0.413	0.1198	35.97%
Length of training sessions (min)	18	0.999 (0.96 to 1.04)	0.958	0.1746	44.90%
Number of supervised sessions	18	0.98 (0.97 to 0.99)	0.002	0.01159	5.22%
End of intervention (gestational week)	18	0.95 (0.81 to 1.12)	0.557	0.1399	40.24%

Explanation: OR—odds ratio; 95% CI—95% confidence interval; I^2^—describes the relative heterogeneity (inconsistency); τ^2^—describes the absolute heterogeneity (between-study variance).

**Table 4 ijerph-20-06069-t004:** Evaluation of covariates for the effect of exercise during pregnancy on preeclampsia.

	k	OR (95% CI)	*p*-Value	τ^2^	I^2^
Overall effect	9	0.62 (0.38 to 1.02)		0.1347	22.66%
Mean age	9	0.96 (0.77 to 1.19)	0.71	0.2347	33.10%
Mean BMI	8	1.14 (0.89 to 1.45)	0.30	5.8 × 10^−7^	0.0%
Duration of intervention (weeks)	9	0.97 (0.87 to 1.08)	0.63	0.2055	28.96%
Training sessions per week	9	0.73 (0.50 to 1.06)	0.10	0.07024	13.09%
Length of training sessions (min)	9	0.98 (0.94 to 1.02)	0.35	0.2829	36.65%
Number of supervised sessions	9	1.01 (0.99 to 1.02)	0.53	0.1711	27.46%
End of intervention (gestational week)	9	0.92 (0.75 to 1.13)	0.45	0.6574	12.08%

Explanation: OR—odds ratio; I^2^—describes the relative heterogeneity (inconsistency); τ^2^—describes the absolute heterogeneity (between-study variance).

**Table 5 ijerph-20-06069-t005:** Summary of findings.

Exercise Compared to Control Group for Reducing GDM, Preeclampsia, and Withdrawal						
Patient or Population: Pregnant Healthy Women Setting: Various Settings Intervention: Exercise Comparison: The Comparators Were No Exercise, Usual Care, Advice to Stay Active, or Other Active Comparative Interventions (as Listed above). Studies with Co-Interventions such a Dietary Intervention/Advice Were Included.Usual Care						
Outcomes	Anticipated Absolute Effects * (95% CI)		Relative Effect (95% CI)	№ of Participants (Studies)	Certainty of the Evidence (GRADE)	Comments
	Risk with Usual Care	Risk with Exercise				
GDM	107 per 1000	70 per 1000 (53 to 92)	RR 0.66 (0.50 to 0.86)	5585 (18 RCTs)	⨁⨁⨁ Moderate ^a,b,c^	Exercise likely results in a large reduction in GDM.
Preeclampsia	53 per 1000	34 per 1000 (22 to 54)	RR 0.65 (0.42 to 1.03)	2861 (9 RCTs)	⨁⨁ Low ^d^	Exercise may reduce/have little to no effect on preeclampsia, but the evidence is uncertain.
Withdrawal	123 per 1000	120 per 1000 (104 to 136)	RR 0.97 (0.84 to 1.10)	6767 (20 RCTs)	⨁⨁⨁⨁ High	Exercise results in no difference in withdrawal.
GRADE Working Group grades of evidence High certainty: We are very confident that the true effect lies close to that of the estimate of the effect. Moderate certainty: We are moderately confident in the effect estimate: the true effect is likely to be close to the estimate of the effect, but there is a possibility that it is substantially different. Low certainty: Our confidence in the effect estimate is limited: the true effect may be substantially different from the estimate of the effect. Very low certainty: We have very little confidence in the effect estimate: the true effect is likely to be substantially different from the estimate of effect.						

* The risk in the intervention group (and its 95% confidence interval) is based on the assumed risk in the comparison group and the relative effect of the intervention (and its 95% CI). CI—confidence interval; RR—risk ratio. Explanations: ^a^ There is a moderate I^2^ of 47.68%, and a little variability of the point estimates, with 15 studies in favor of the intervention group and 3 in favor of the control. ^b^ The CI differs from 0.50 to 0.87, but do not cross 1 and become harmful. ^c^ Downgraded because of high probability of publication bias–sensibility analysis shows a lower non-significant effect when removing the studies from the same researcher group. ^d^ The domain is downgraded twice because the confidence intervals include both appreciable benefit and appreciable harm. The number of “⨁” shows the grade of evidens—one cross is very low, two is low and so on.

## Data Availability

Not applicable.

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
