# Peer review of "The Effects of Exercise during Pregnancy on Gestational Diabetes Mellitus, Preeclampsia, and Spontaneous Abortion among Healthy Women—A Systematic Review and Meta-Analysis"

_ijerph, 2023, doi:10.3390/ijerph20126069_

Round 1

Reviewer 1 Report

Thank you for the opportunity to review this interesting work. The authors have aimed to perform a systematic review and meta-analysis to compare the effects of different exercise modalities on the risk of GDM, preeclampsia, spontaneous abortion, withdrawal and adverse events in healthy pregnant women. The main findings suggested that exercise during pregnancy was beneficial in the prevention of GDM.

Overall, the paper is well done and the authors have addressed pregnancy complications of concern. Since certain levels of exercise are recommended for healthy pregnancies without any potential complications, the findings from this meta-analyis might help guide future recommendations for exercise during pregnancy. Tables and figures are well done. It was interesting to see the different sub group analysis. Although it was surprising that no studies reported on spontaneous abortion.

Minor Comments

·       The results say that “20 studies” were included but the abstract only mentions “18 studies”. Please double check.

·       Lines 84-85: probably cite the reference for the latest PRISMA guidelines unless the protocol was developed using the previous guidelines.

“Page M J, McKenzie J E, Bossuyt P M, Boutron I, Hoffmann T C, Mulrow C D et al. The PRISMA 2020 statement: an updated guideline for reporting systematic reviews BMJ 2021; 372 :n71 doi:10.1136/bmj.n71 “

·       Line 101: which adverse outcomes were examined? Probably mention somewhere in the results.

·       Line 124-125: what kind of dietary intervention were included?

·       Line 128: should list the gestational age at which GDM was diagnosed or was it not included in the definition.

·       Line 129-130: preeclampsia is usually defined as new onset of hypertension after 20 weeks gestation. Should mention the gestational age in the definition.

·       Line 243: what does “less extensive” intensity refer to?

·       Figure 1: please double check the numbers in the flowchart.

·       Table 2: Probably can remove the column on “conflict of interest”. Maybe include as a footnote.

·       Please list some directions for future work and comment on generalizability.
